# Comparing the Effect of HPP on the Structure and Stability of Soluble and Membrane-Bound Polyphenol Oxidase from ‘Lijiang Snow’ Peach: Multispectroscopic and Molecular Dynamics Simulation

**DOI:** 10.3390/foods12091820

**Published:** 2023-04-27

**Authors:** Hengle Zhou, Shenke Bie, Zi Li, Linyan Zhou

**Affiliations:** Faculty of Food Science and Engineering, Kunming University of Science and Technology, Kunming 650500, China

**Keywords:** polyphenol oxidase, high-pressure processing, conformational changes, molecular docking, high-pressure simulation

## Abstract

Polyphenol oxidase (PPO) easily causes fruits and vegetables to lose their color and nutritional value. As a non-thermal process, high-pressure processing (HPP) showed different inactivation effects on endogenous enzymes. In this work, soluble PPO (sPPO) and membrane-bound PPO (mPPO) from ‘Lijiang snow’ peaches were purified, and then the effect of high pressure on the conformation of sPPO and mPPO was investigated and compared at the molecular level. The maximum activation of sPPO and mPPO by 11.2% and 4.8% was observed after HPP at 200 MPa, while their activities both gradually decreased at 400 MPa and 600 MPa; in particular, the residual activities of sPPO and mPPO at 600 MPa for 50 min were 41.42% and 72.95%, respectively. The spectroscopic results indicated that the secondary structure of PPOs was little affected by HPP, but HPP led to obvious changes in their tertiary structure. The simulations showed that the decreasing distance between the copper ion and His residue in the copper-binding region of two PPOs at 200 MPa was favorable to catalytic activity, while the increasing distance between copper ions and His residues and the disordered movement of the loop region above 400 MPa were unfavorable. In addition, the structure of sPPO was relatively looser than that of mPPO, and high pressure showed a more significant effect on the conformation of sPPO than that of mPPO. This study clarified the effect of HPP on PPO’s structure and the relationship between its structure and activity and provided a basis for the prevention of enzymatic browning.

## 1. Introduction

Polyphenol oxidase (PPO) is a very common metalloenzyme that oxidizes phenolic substances and widely exists in nature [1]. It was reported that the copper-binding region plays a crucial role in the catalytic reaction of the PPO, where each copper ion binds to three His residue ligands to form an active center [2]. PPO can cause enzymatic browning of fruits and vegetables, resulting in deterioration of their color and nutritional composition, which can seriously shorten their storage time, and even reduce their commercial value. It was reported that more than half of the losses in fruit and vegetable products in the world were caused by enzymatic browning [3]. Therefore, using certain methods to control enzymatic browning is an essential measure in the process of the fruit and vegetable industry. 

Moreover, PPO exists in various forms in plants, among which the two main forms reported are sPPO and mPPO [4]. It was reported that sPPO was mainly located in the plastid of cells, while mPPO bound strongly or weakly to organelle membranes, and they could be interconverted with each other under specific conditions [4]. Moreover, the catalytic oxidation modes of the two PPOs were also different. The catalytic reaction process between sPPO and substrate was slow, while mPPO could immediately bind to the substrate due to its powerful enzymatic activity, resulting in a rapid color change [5]. Previous studies have reported that there were some differences in enzyme properties between sPPO and mPPO. Jia et al. [6] compared the effects of inhibitors on sPPO and mPPO in peach fruit, and the results showed that L-cysteine and ascorbic acid have a stronger inhibitory effect on sPPO than mPPO. Therefore, we summarized the following points: (i) mPPO’s activity was usually higher than that of sPPO; (ii) there are certain differences in the optimum substrate, pH, and temperature, it was found that mPPO was generally more resistant to acid/alkaline and pressure/temperature than sPPO; (iii) sPPO and mPPO could show different tolerance in the presence of inhibitors, and their inhibition mechanisms were also different, with mPPO typically exhibiting stronger inhibition tolerance generally [7,8,9,10,11,12]. The differences in the enzyme properties were mainly attributed to their structure, which could be determined by analyzing the amino acid sequences of the enzyme, and other information, such as the enzyme’s bioinformatics, function, and evolution, could be obtained. 

As a non-thermal processing technology, high-pressure processing (HPP) can transmit pressure through water to achieve the desired effect of sterilization and inactivation of endogenous enzymes. Compared to thermal processing, HPP can rupture the food tissue cells and inactivate some proteases at room temperature, and it can maintain the sensory and nutritional qualities of food [13]. In recent years, advanced thermal technologies, such as microwave heating, ohmic heating, and radiofrequency heating, have also been widely studied [14,15]. However, these techniques to inhibit PPO activity still rely mainly on the thermal effect, which more or less destroys the sensory and nutritional quality of fruit and vegetables. Therefore, non-thermal technologies, such as HPP, pulsed electric field (PEF), and high-pressure homogenization (HPH), are still a hot spot in fruit and vegetable processing. Numerous studies showed that HPP exhibited different inactivation effects on PPO activity depending on the resource, and its activity was even activated at lower pressure [16,17,18,19,20,21]. The activation of PPO activity is possibly due to HPP causing the active center of PPO to be gradually exposed to the protein’s surface, making contact between PPO and the substrate easier [22]. In contrast, higher pressure might lead to changes in the secondary and tertiary structures of PPO, resulting in folding and unfolding of the polypeptide chain of the enzyme, and finally irreversible destroy the three-dimensional conformation and directly affect the catalytic activity of the enzyme [23,24]. However, the current studies reported on the effects of HPP on the activity and conformation of PPO did not clarify the forms of PPO, limiting the further exploration of the inactivation mechanism of PPO. Wang et al. [10] reported that the activities of sPPO and mPPO in ‘Lijiang snow’ peach after HPP at 550 MPa were decreased by 89% and 12%, respectively, indicating that mPPO exhibited more resistance to pressure than sPPO. Zhou et al. [25] also reported that the residual activity of mPPO and sPPO in pear was decreased to 59.6% and 13.5% by HPP (600 MPa, 16 min), respectively. Thus, the exploration of the effect of HPP on the conformation of different forms of PPOs is necessary to improve the application of HPP to inhibit enzymatic browning.

The wide range of PPO resources and the different forms of PPO both lead to the different inhibition effects and mechanisms of PPOs exhibited by the same technique. At present, although numerous studies have investigated the inhibition of PPO activity by HPP, most of the references focused only on the inhibition effect, and few studies explored the difference in the inhibition mechanisms between different forms of PPO at the molecular level. Therefore, in this work, we analyzed the bioinformatics and three-dimensional structures of sPPO and mPPO purified from ‘Lijiang snow’ peaches, investigated the conformational changes of sPPO and mPPO treated by HPP, and further explored the relationship between enzymatic activities and their structure at the molecular level through molecular dynamics simulations combined with multispectral techniques. This study can help to better elucidate the inactivation mechanism of HPP on the structure and stability of sPPO and mPPO, and it can provide a basis for the reduction in enzymatic browning during the processing and storage of peach products. 

## 2. Materials and Methods

### 2.1. Materials

In this work, ‘Lijiang snow’ peaches were purchased from a plantation in Lijiang, Yunnan Province. In addition, the samples and materials required for this work were similar to those in our previous study [26]. 

### 2.2. Extraction and Purification of sPPO and mPPO

Extraction and purification of sPPO and mPPO from ‘Lijiang snow’ peaches were performed as described in a previous study [26]. Briefly, a 1:1 (*w*/*v*) ratio of peaches and phosphate buffer (pH 6.8, 0.05 mol/L) was homogenized and centrifuged to obtain the crude sPPO and mPPO extract, and then ammonium sulfate fractional precipitation and dialysis were performed. Finally, sPPO was purified by DEAE-Sepharose fast flow column chromatography using a chromatography system (Bio-Rad NGC, Hercules, CA, USA), and mPPO was purified by DEAE-Sepharose fast flow and Sephacryl S-200 HR 16/60 flow column chromatography. SDS-PAGE and Native-PAGE were used to evaluate the purity of sPPO and mPPO according to the method described by Laemmli [27].

### 2.3. Enzyme Assay

According to Terefe et al. [28], 2.5 mL pH 6.8 0.05 mol/L phosphate buffer rich in 0.2 mol/L catechol solution and 0.5 mL PPO formed a reaction system. The absorbance changes of the reaction system within one minute were measured at 420 nm using a UV-Vis spectrophotometer with two beams (T9CS, Beijing Percy General Instrument Co., Ltd., Beijing, China). The residual enzyme activity (*R_A_*) was the percentage of enzyme activity after HPP compared to untreated enzyme activity, which was used to indicate the change in PPO activity.

### 2.4. Bioinformatics Analysis

The nESI-LC-MS/MS of sPPO and mPPO was performed by a third-party company (BGI Genomics Co., Ltd. Shenzhen, China), as described previously [26]. The amino acid composition and biological properties of sPPO and mPPO were analyzed using the ProtParam program (as shown in Appendix A); the hydrophilicity and hydrophobicity were analyzed using the ProtScale program; the secondary structures were predicted by the GOR Ⅳ program; the subcellular localization was predicted using the Hum-mPLoc 3.0 program.

### 2.5. Three-Dimensional Structure Modeling

Three-dimensional structures of sPPO and mPPO were built using the Swiss model to screen protein models with high amino acid sequence homology to PPOs as templates. The protein models with the highest homology with sPPO and mPPO were downloaded from the PDB database, multiple three-dimensional models were constructed using SWISS-Model, and the one with the highest score was retained. Finally, Ramachandran plots were used to assess whether the conformation of each amino acid in the constructed three-dimensional structures of PPOs was correct.

### 2.6. High-Pressure Processing

In this experiment, a 10 L high-pressure instrument (XC-LF3AH, Jiangmen Xiecheng Machinery Co., Ltd., Jiangmen, China) was used to treat sPPO and mPPO. sPPO and mPPO were treated by HPP at 200, 400, and 600 MPa for 10, 20, 30, 40, and 50 min. 

### 2.7. CD Spectra Analysis

The CD spectra were determined using a CD spectrophotometer (Bio-Logic MOS-450, Grenoble, France), according to the method reported by Yi et al. [24]. The wavelength range of the scan was set to 190–250 nm, the scan rate was set to 5 nm/s, the bandwidth was set to 1 nm, the resolution was set to 0.2 nm, and the response time was set to 0.25 s. 

### 2.8. Fluorescence Spectra Analysis

The fluorescence spectra were measured using a fluorescence spectrometer (Hitachi F-4500, Tokyo, Japan), according to the method reported by Chen et al. [29]. The excitation wavelength was set to 280 nm, and the emission spectrum was scanned in the range of 300–500 nm. Simultaneously, the ANS-binding fluorescence of sPPO and mPPO was also studied. The excitation wavelength was set to 390 nm, and its emission spectrum was scanned in the range of 420–600 nm. 

### 2.9. Molecular Dynamics Simulation

Molecular dynamics (MD) simulation of sPPO and mPPO under high pressure (0.1, 200, 400, and 600 MPa) was performed using the Gromacs software (version 2019; University of Groningen, Groningen, the Netherlands) [30]. All simulations were similar to those in our previous study [22] and were performed for 100 ns of all-atom molecular dynamics. After the entire simulation process was completed, the result files were extracted through the trajectory.

### 2.10. Molecular Docking

Molecular docking after each pressure simulation was performed using SYBYL 2.1.1 software (Tripos, Inc., St. Louis, MI, USA) to explore the binding mechanism of sPPO and mPPO with catechol (PubChem CID:289). The interaction results were visualized using PyMOL.

### 2.11. Statistical Analysis

All experimental measurements were repeated 3 times. The results of PPO activities and spectroscopy experiments were analyzed and plotted using Origin 8.5 software (Origin Labs, Northampton, MA, USA).

## 3. Results and Discussion

### 3.1. Bioinformatics and Three-Dimensional Structure Analysis of sPPO and mPPO

The purification results of sPPO and mPPO from ‘Lijiang snow’ peaches are shown in Table 1. It could be seen that the specific activity of the sPPO and mPPO was 18,701.59 and 96,112.16 U/mg, respectively. It could be seen that the specific activity of mPPO from ‘Lijiang snow’ peaches was 5 times that of sPPO. Previous studies have also found that the specific activity of PPO varied greatly depending on its source [7,31,32].

The bioinformatics analysis of sPPO and mPPO was carried out by prediction programs as shown in Appendix A and Appendix A. Results showed that sPPO and mPPO had the same amino acid composition, but their proportions were slightly different. Both sPPO and mPPO were stable hydrophilic proteins, but they had different locations. The location of PPO in the cell tissues of fruits and vegetables varies depending on the species, variety, and maturity. It was generally believed that sPPO mainly existed in the cell fluid, while mPPO mainly existed in the organelles such as chloroplasts and mitochondria [33]. As expected, sPPO was presented in the cytoplasm without a transmembrane helix, whereas mPPO located in mitochondria contained two transmembrane helices and a membrane integrin. Liu et al. [7] also found a transmembrane helix existed in the Fuji apple mPPO. The transmembrane region is an important protein channel connecting the extracellular and intramembrane environments; it can carry out various activations and reactions to regulate the morphological and functional changes inside and outside the membrane [34]. Thus, it speculated that mPPO would be subjected to stress reaction, pass through the inner membrane of mitochondria, and then be converted into sPPO and participate in the enzymatic browning reaction when the tissue cells of peach are subjected to mechanical damage, collisions, and other adverse conditions. In general, the differences that existed in the bioinformatics of the two PPOs, including subcellular localization and transmembrane structure, could indicate that their structure and function will be different [35]. 

Homologous modeling results showed that PPO (PDB ID: 6ELS) in Malus domestica had 57% similarity with sPPO, while PPO (PDB ID: 4Z11) in Coreopsis grandiflora showed 47% similarity with mPPO. Therefore, these crystal structure models were used as templates for the homology modeling of the PPOs. The quality of models was checked by PROCHECK as shown in Figure 1a,b; 94.5% and 94.4% of residues of sPPO and mPPO were distributed in the allowable region, respectively, indicating that they both have a good quality model. In addition, 87.6% and 89.0% of amino acids in sPPO and mPPO were located in the core region (red region), respectively, which might indicate that the conformation of sPPO was more unstable than that of mPPO. As shown in Figure 1c,d, the advanced structure of sPPO from ‘Lijiang snow’ peaches was mainly composed of four α-helices, two β-sheets, and multiple random coils, while that of mPPO mainly consisted of six α-helices, eight β-sheets, and multiple random coils. Meanwhile, it could be seen that the conformation of sPPO was relatively loose, while the conformation of mPPO was more compact. In addition, it was found that mPPO has more His residues in the activity region than sPPO, which might be the reason for the higher catalytic activity of mPPO [1].

### 3.2. Effect of HPP on the Activity of sPPO and mPPO

As can be seen from Figure 2a, the activity of sPPO was activated after HPP at 200 MPa; in particular, its activity was increased by 11.2% after HPP for 20 min, and then the activity continued to decrease when the treatment time was prolonged. When the pressure reached 600 MPa, sPPO activity decreased significantly after treatment for 10 min, and its *R_A_* was decreased to 41.42% after treatment for 50 min. These results were similar to those of our previous study on mushroom PPO [27]. As shown in Figure 2b, a smaller activation of mPPO activity was observed after HPP, and its activity was only activated by 4.8% and 3.8% after HPP at 200 MPa for 10 min and 20 min, respectively. Meanwhile, mPPO activity gradually decreased with the increase in treatment time under 400 MPa. However, the *R_A_* of mPPO was still beyond 72.95% even when the pressure was increased to 600 MPa, indicating that mPPO was more resistant to pressure. Zhou et al. [25] also found that the *R_A_* of mPPO from pear after HPP (600 MPa, 16 min) was 59.6%, which was significantly higher than that of sPPO (13.5%). It could be seen that mPPO was more active than sPPO under high pressure. 

### 3.3. Multispectral Analysis

As shown in Figure 3a, the contents of α-helix, β-sheet, β-turn, and random coils of native sPPO were 22.95%, 16.01%, 20.22%, and 40.82%, which all fluctuated slightly after HPP. Similar changes were observed for the mPPO’s secondary structure with initial α-helix content of 21.73%, β-sheet content of 14.23%, β-turn content of 22.35%, and random coils content of 41.69%. It could be seen that the secondary structures of sPPO and mPPO were both stable in response to HPP with a pressure range of 200–600 MPa. This result was in accordance with previous studies indicating that the secondary structure of PPO was generally not obviously changed after HPP [26,36]. Yi et al. [24] also reported that the secondary structure of mushroom PPO was relatively stable even at 800 MPa.

The changes in endogenous fluorescence can reflect the influence of HPP on the tertiary structure of PPO. As shown in Figure 3b, it could be seen that the fluorescence intensities of sPPO and mPPO both gradually increased; in particular, the fluorescence intensities of sPPO and mPPO after HPP at 600 MPa for 50 min were increased by 8.1% and 15.2%. The increases in fluorescence intensities showed that HPP could cause the microenvironment of Trp residues to change and the tertiary structure of PPO to be destroyed. Hou et al. [37] found that the endogenous fluorescence intensity of SOD was increased by 5.7% after HPP at 500 MPa for 20 min. HPP could lead to the expansion of the protein peptide chain, and then Trp residues in the internal nonpolar environment would be more exposed to the polar environment, resulting in an increase in fluorescence intensity [38]. In addition, the activity of mPPO changed less than that of sPPO, while its fluorescence intensity increased more. It was found that the fluorescence intensity of a protein is related to fluorescence quantum yield, extinction coefficient, and substance content [39]. As shown in Appendix A, the extinction coefficient of mPPO was 105240, higher than that of sPPO (100045), which meant that the mPPO’s fluorescence produced by chromophores such as Trp residues was brighter, and thereby the change in fluorescence intensity was more pronounced.

Exogenous fluorescence is usually used to characterize the changes in protein hydrophobicity. The fluorescence intensity of sPPO first decreased after HPP treatment at 0.1–400 MPa and then increased at 600 MPa, but its value did not exceed the fluorescence intensity of native sPPO (Figure 3c). It could be seen that the activity of sPPO after HPP was consistent with the change in its fluorescence intensity. The decrease in hydrophobicity of sPPO after HPP under lower pressure might be explained by HPP being able to disrupt the complete structure of the protein, and some amino acid residues tend to move toward the internal hydrophobic environment [40]. Under higher pressure, HPP could promote the solvation of proteins and make more hydrophobic regions contact solvents [41]. However, different changes in fluorescence intensities were observed for mPPO after HPP. The fluorescence intensity of mPPO gradually increased; in particular, the fluorescence intensity was increased by 32.9% after HPP at 600 MPa for 50 min. Li et al. [42] reported that the fluorescence intensity of inulin transferase was increased by 18.8% after HPP at 600 MPa. HPP could destroy the tertiary structure of the protein, which made the hydrophobic side chains in the natural protein state gradually buried into the inside of the protein, resulting in more residues exposed to the protein surface, enhancing the protein’s surface hydrophobicity [43].

### 3.4. MD Simulations of sPPO and mPPO at the Single-Molecule Level

#### 3.4.1. RMSD and RMSF Analysis of sPPO and mPPO

As shown in Appendix A, the RMSD of sPPO fluctuated more significantly than that of mPPO, and the stable value of RMSD of sPPO was greater than that of mPPO, indicating high pressure showed greater influences on the conformation of sPPO than that of mPPO. 

Likewise, it could be seen that RMSF of sPPO fluctuated more strongly at the pressures of 400 MPa and 600 MPa than at 0.1 MPa and 200 MPa, indicating that 400 MPa and 600 MPa were more likely to cause the instability of amino acid residues. In addition, the RMSF value of Res250 (mainly located in the loop region) in sPPO increased significantly at all simulated pressures, suggesting that high pressure could lead to a significant movement of the loop region of sPPO. In contrast, as compared with sPPO, the RMSF of mPPO fluctuated slightly under all simulated pressures, also indicating that high pressure had less significant effects on mPPO’s conformation. 

#### 3.4.2. The Secondary and Tertiary Structure Analysis of sPPO and mPPO

As can be seen from Figure 4, the secondary structures of sPPO and mPPO under the pressure of 200 MPa–600 MPa were stable as compared to 0.1 MPa; in particular, the α-helix contents of sPPO and mPPO were both kept stable. Zhou et al. [27] also found that the α-helix and β-sheet of the mushroom PPO simulated under high pressure remained unchanged at 600 MPa. The α-helical composition in the active centers of sPPO and mPPO remained stable at 200–600 MPa, indicating that the secondary structures of sPPO and mPPO showed strong stability under high pressure. These results also confirmed the results of the CD spectrum.

The surface solvent accessible area (SASA) is used to describe the hydrophobicity and to characterize the change in the tertiary structure of a protein. As shown in Table 2, the hydrophobic SASA of sPPO gradually decreased from 12,113.271 nm^2^ (0.1 MPa) to 11,603.713 nm^2^ (400 MPa) and then increased to 12,109.229 nm^2^ at 600 MPa, indicating that high pressure caused the hydrophobicity of sPPO to decrease firstly under lower pressure and then increase under higher pressure. The hydrophobic SASA value of mPPO gradually increased with increasing pressure, indicating that high pressure led to an increase in the hydrophobicity of mPPO. Those changes in hydrophobic SASA were consistent with the results of exogenous fluorescence of sPPO and mPPO treated by HPP. It could be seen that the hydrophobic changes of sPPO and mPPO under high pressure were different, which might be due to the hydrophobic cavity of sPPO not being heavily surrounded by random coils and being more exposed to the outside. Low pressure caused the hydrophobic cavity to be buried, while higher pressure caused the polypeptide chain to rearrange and resulted in more hydrophobic residues exposed to the protein surface [44]. The 3D structure of mPPO was relatively compact, and thus it was deduced that low pressure had resulted in no obvious changes in the hydrophobic cavity, while high pressure could lead to the rearrangement of polypeptide chains and the increase in fluorescence intensity. 

#### 3.4.3. The Copper-Binding Region Analysis of sPPO and mPPO

The copper-binding region is a very key component of the PPO structure and is crucial for the catalytic reaction of PPO [1]. Figure 5 shows the changes in copper-binding regions of sPPO and mPPO at all simulated pressures. Compared with 0.1 MPa, the distance between the two copper ions in the copper-binding region of sPPO changed from 4.1 Å to 3.2 Å, 3.6 Å, and 9.6 Å. Similarly, the distance between the two copper ions of mPPO after simulation had the same trend as that of sPPO. Therefore, it is obvious that the copper ions of PPO’s copper-binding regions were easily wrecked under high pressure. Meanwhile, it could be seen that the distance between the two copper ions of sPPO was more easily influenced by high pressure than that of mPPO. The reduction in the distance between copper ions was conducive to the catalytic reaction of PPO, while the increase in the distance would lead to a decrease in PPO’s activity [45]. The changes in activities of sPPO and mPPO were consistent with the above statement.

On the other hand, the catalytic function of PPO’s copper-binding region was also affected by the surrounding His residues around the copper ions. The distance between the copper ion and His 15 of sPPO was 2.5 Å at 0.1 MPa, and this value decreased by 0.7 Å at 200 MPa, and then the distance increased gradually under 400 MPa and 600 MPa. In particular, the distance between the copper ion and His 146 rose from 2.0 Å (200 MPa) to 5.0 Å (600 MPa), indicating that the stability of the copper-binding region of sPPO was destroyed by higher pressure. Similar changes were observed in the copper-binding region of mPPO. The reduction in the distance between copper ions and His residues favored the catalytic activity of PPO [46]. Moreover, the changes in the distance between the Cu ions and His residues of sPPO were greater than those of mPPO after simulation at 600 MPa, which was also consistent with the changes in PPO’s activities. The proper folding of the polypeptide chain in the copper-binding region plays a key role in maintaining PPO activity, and slight changes can disrupt the catalytic activity of copper. In general, it could be seen that the copper-binding region of sPPO changed more under simulation at a higher pressure than mPPO, and the change in the binuclear copper-binding region of PPO might be the main reason for the different inactivation effects of HPP on sPPO and mPPO. 

#### 3.4.4. The Three-Dimensional Conformational Change Analysis of sPPO and mPPO

It could be seen that the volume of both sPPO and mPPO decreased gradually with the increase in simulated pressure (Table 2). The radius of gyration (R_g_) can indicate the tightness of a protein’s structure, and the larger its value is, the looser the structure is. The R_g_ values of sPPO and mPPO both gradually decreased with increasing pressure, indicating that high pressure destroyed the ordered structure of PPOs. High pressure could force water molecules into the protein hydrophobic cavity, which causes the destruction of the ordered structure and compression of the protein cavity, thus reducing the volume of the protein [47,48]. 

The stability of protein structure is mainly determined by the internal forces of the protein molecule; in particular, the hydrogen bonds are the main force maintaining the protein’s structure [49]. The number of hydrogen bonds within the sPPOs decreased from 186 to 171 with the increase in simulated pressure. Our previous study showed a similar result: the number of hydrogen bonds of mushroom PPO changed from 302 to 276 with the increase in simulated pressure [27]. Meanwhile, the number of hydrogen bonds within mPPO fluctuated slightly with the increase in simulated pressure, and the number of hydrogen bonds remained around 406 ± 5. It was deduced that sPPO had a soft structure with weaker intermolecular forces as compared to mPPO, and higher pressure would easily destroy the structure of sPPO, leading to a significant reduction in enzyme activity. However, mPPO had more hydrogen bonds internally, which could better maintain the overall stability of protein structure under high pressure. 

The loop regions of PPO consisted of many random coils with high specificity, which were usually closely related to their catalytic activity [50]. As shown in Figure 6a, when the simulation pressure gradually increased, the arrows in the loop region increased in number and length. In particular, some arrows appeared in the main chain of sPPO stimulated under pressure at 600 MPa, which showed that high pressure had a serious impact on sPPO’s conformation. However, a different situation was observed for the movement of mPPO’s amino acid residues, as shown in Figure 6b. There were only a few short arrows that appeared in mPPO even under high-pressure simulation, indicating that the 3D structure of mPPO was not easily disrupted by high pressure. The difference in the effects of high pressure on the 3D conformations of sPPO and mPPO was also consistent with the changes in RMSD and RMSF values. 

Obviously, high pressure could lead to different effects on the two PPO conformations. Firstly, the helical structure of sPPO was shorter as compared to that of mPPO, and the loop region was far away from the main chain, which weakened the anchoring effect of the main chain on the loop region, and the flexibility of the loop region was enhanced; thereby, high pressure could cause a significant effect on the loop region [51]. In contrast, mPPO had a stable and tightened structure composed of a longer α-helix, multi-strand β-sheets, and random coils, which could increase the resistance of mPPO to high pressure. In addition, sPPO had an unknown functional region, which was mainly located in the loop region (Appendix A). It was deduced that the unknown functional region contributed to the strong movement of the loop region caused by high pressure. In the case of mPPO, the transmembrane helix of transmembrane protein could connect the loops and peptide chains in the region before and after the transmembrane region, making the binding between amino acid residues more stable and thus better maintaining the stability of the protein structure [52]. Schumacher et al. [53] found that some thermophilic bacteria contained transmembrane helices, which could make the internal binding of molecules more stable and help to form the structural basis for stable activity in extreme environments. As mPPO is a membrane integrator protein, this might be one of the key factors why it has a higher resistance to high pressure. 

### 3.5. Molecular Docking

The affinity changes of sPPO and mPPO after high-pressure simulation with substrates were evaluated by molecular docking. Table 3 shows that both sPPO and mPPO after 200 MPa simulation had the highest T-score, which was consistent with the activity changes of PPOs after HPP.

The hydrogen bonding between molecules could play an important role in substrate–enzyme interactions [54]. As shown in Figure 7a, sPPO under 0.1 MPa formed three hydrogen bonds with catechol with 2.53 Å average distance, while three hydrogen bonds with 2.3 Å average distance were formed when sPPO was simulated at 200 MPa. More or shorter hydrogen bonds between PPO and catechol signified tighter binding and stronger interactions. When the simulated pressure for sPPO was increased to 400 MPa and 600 MPa, there were only two hydrogen bonds formed with catechol, which was consistent with the changes in sPPO activity after HPP at 400–600 MPa. There were similar changes in hydrogen bonds between catechol and mPPO (Figure 7b). These results indicated that the site of the binding of catechol with PPOs changed greatly under different high pressures. Huang et al. [55] reported that high pressure promoted the interaction between β-lactoglobulin and EGCG, and there were five hydrogen bonds formed under 600 MPa, more than the four hydrogen bonds under 0.1 MPa. In addition, as shown in Figure 7c,d, the amino acid residues involved in hydrophobic interactions changed with increasing pressure. The docking results also verified the above findings that the hydrophobic interaction between PPOs and substrate was increased under high pressures, and the binding pattern and environment of PPOs also changed considerably, leading to a change in binding energy.

## 4. Conclusions

In this work, the activation of sPPO and mPPO occurred after HPP at 200 MPa, while the activity gradually decreased at 400–600 MPa. Moreover, the inactivation effect of HPP on sPPO was significantly stronger than that on mPPO. MD simulations showed that these changes in the copper-binding region of PPOs at 200 MPa were favorable to the catalytic activity, especially the decreases in distance between copper ion and His residue, while those changes of PPOs above 400 MPa, including the increasing distance between copper ions and His residues and the disordered movement of loop region, could lead to a decrease in catalytic activity. Moreover, the changes in the conformation of sPPO under high pressure were much greater than those of mPPO. This was mainly due to the structure of sPPO being relatively loose as compared to mPPO, and the loop region of sPPO being too far away from the main chain, which weakened the anchoring effect of the main chain on the loop region, thus leading to a significant destabilization effect of high pressure on the loop region of sPPO. In addition, mPPO is a membrane-integrated protein, and the outer and inner membrane structures could be closely connected by a transmembrane helix, which made it form a highly compact elliptical structure. Therefore, the three-dimensional structure of mPPO was not damaged obviously under high pressure, which might make it more resistant to high pressure. This study deeply and intuitively explored and compared the inactivation mechanism of HPP on sPPO and mPPO, clarified the correlation between the structural changes of PPOs under high pressure and their activity, and explained the inhibitory effect of HPP on enzymatic browning. The mechanism of HPP on different forms of PPO can provide a great reference for the practical production of HPP in the fruit and vegetable industry.

## Figures and Tables

**Figure 1 foods-12-01820-f001:**
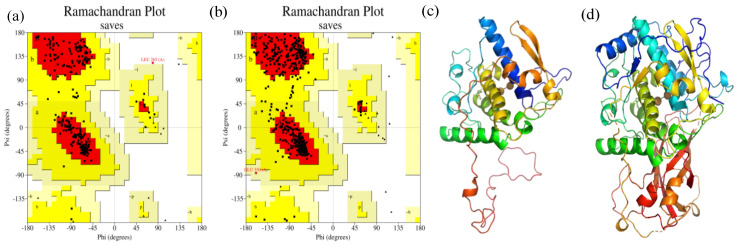
Ramachandran plot of the Psi/Phi distribution of sPPO (**a**) and mPPO (**b**) homology models produced by PROCHECK; the 3D structures of sPPO (**c**) and mPPO (**d**) from Protein Data Bank. The yellow region is the allowable region, the light yellow region is the maximum allowable region, and the other regions are unreasonable amino acid residues that may exist in the three-dimensional conformation.

**Figure 2 foods-12-01820-f002:**
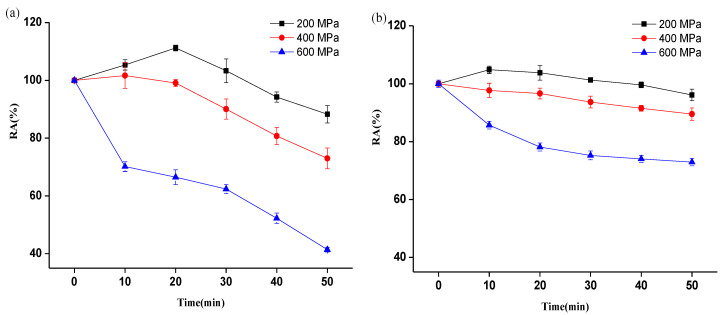
Changes in the activities of sPPO (**a**) and mPPO (**b**) after HPP at 200–600 MPa for 0–50 min. “RA” represents the percentage of PPO’s activity after HPP relative to untreated activity.

**Figure 3 foods-12-01820-f003:**
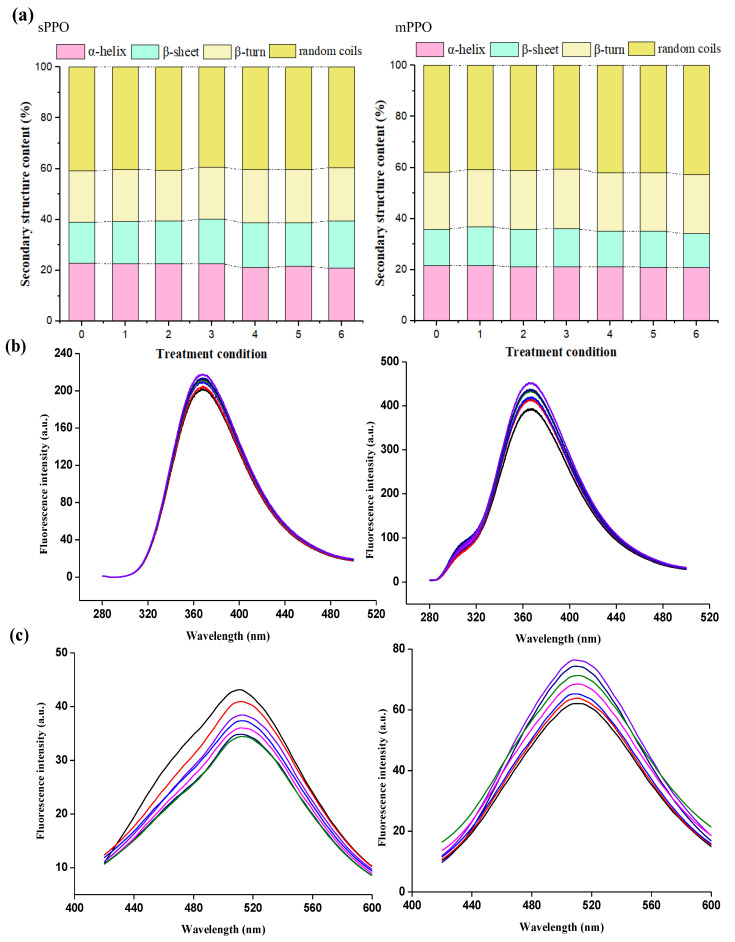
The contents of secondary structure (**a**) of sPPO and mPPO under different HPP conditions (0–6 of the abscissa in the figure represent HPP treatment conditions, which were untreated, 200 MPa–10 min, 200 MPa–50 min, 400 MPa–10 min, 400 MPa–50 min, 600 MPa–10 min and 600 MPa–50 min, respectively); the endogenous fluorescence spectra (**b**) and exogenous fluorescence spectra (**c**) of sPPO and mPPO under different HPP conditions (——: 0.1 MPa; ——: 200 MPa, 10 min; ——: 200 MPa, 50 min; ——: 400 MPa, 10 min;——: 400 MPa, 50 min; ——: 600 MPa, 10 min; ——: 600 MPa, 50 min).

**Figure 4 foods-12-01820-f004:**
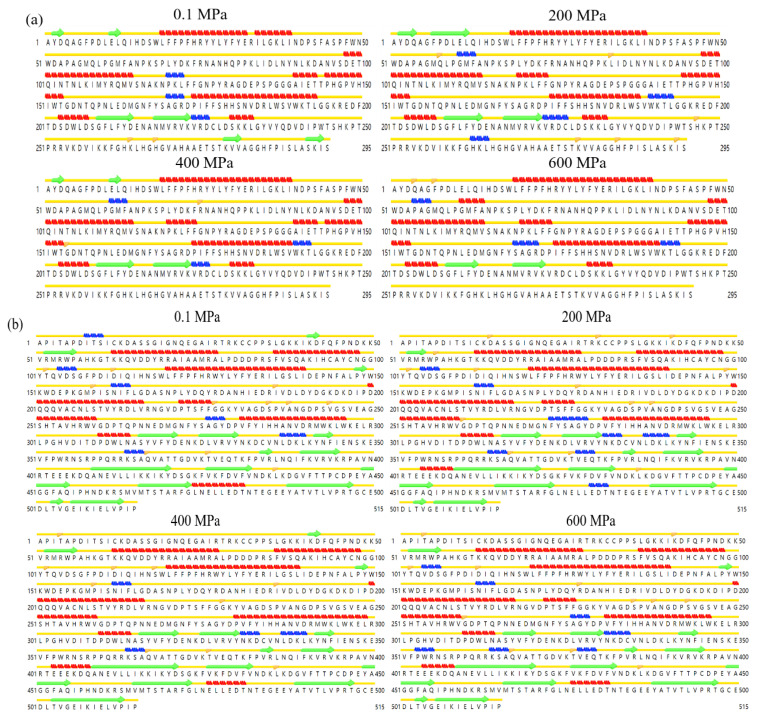
The changes in the secondary structures of sPPO (**a**) and mPPO (**b**) according to MD simulation under different pressure. The letters represent the amino acids that make up the protein (
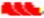
: α-helix; 
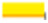
: coil; 
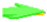
: β-sheet; 
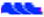
: 3-10 helix; 
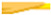
: isolated beta bridge).

**Figure 5 foods-12-01820-f005:**
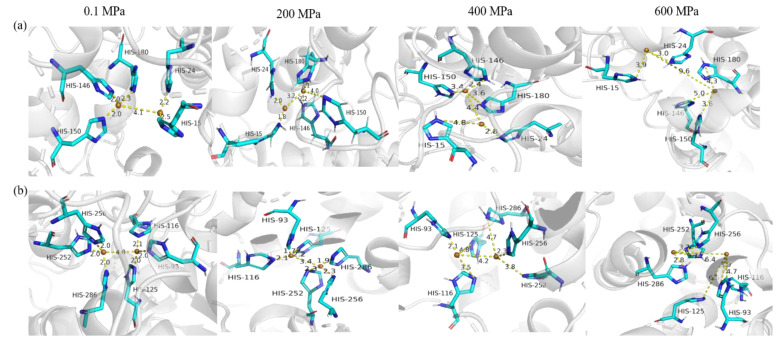
The active center of sPPO and mPPO according to MD simulation under different pressures. (**a**) and (**b**) represent the distance change of copper and His residues in the copper-binding region of sPPO and mPPO, respectively.

**Figure 6 foods-12-01820-f006:**
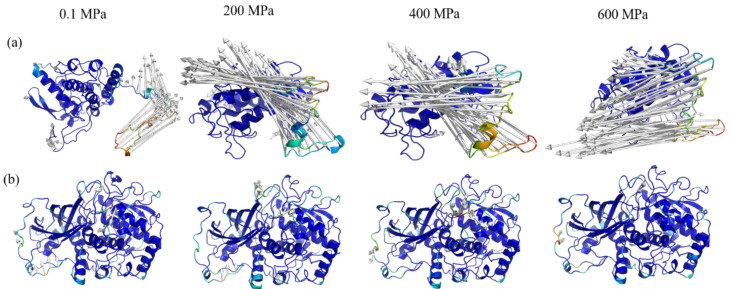
The porcupine diagram of amino acid residue movement of sPPO (**a**) and mPPO (**b**) according to MD simulation under different pressure. The white arrows point from the position of zero at the beginning of the simulation to the position at the end of the simulation.

**Figure 7 foods-12-01820-f007:**
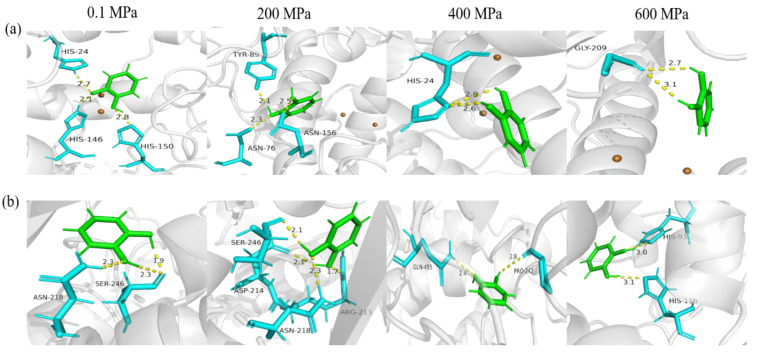
Binding mode of catechol with PPO residues after simulation under different pressure. (**a**,**b**) The 3D view of the binding of sPPO and mPPO with catechol, respectively. Catechol is displayed in green, and the yellow dot line shows the hydrogen bond. (**c**,**d**) The interactions in 2D view. Light green (
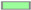
) represents van der Waals force; dark green (
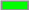
) represents hydrogen bond; other colors represent pi conjugation.

**Table 1 foods-12-01820-t001:** Purification of sPPO and mPPO from ‘Lijiang snow’ peaches.

Purification Stages		Total Activity (U)	Total Protein (mg)	Specific Activity (U/mg)	Yield (%)	Purification Fold
sPPO	Crude extract	106,600	41.82	2549.02	100	1
	40–80% (NH_4_)_2_SO_4_ precipitation	49,650	12.05	4120.33	28.81	1.62
	DEAE Sepharose fast flow	18,350	0.9812	18,701.59	2.35	7.34
mPPO	Crude extract	185,240	53.45	3465.67	100	1
	40–80% (NH_4_)_2_SO_4_ precipitation	106,340	19.06	5579.22	35.66	1.61
	DEAE Sepharose fast flow	86,360	2.188	39,469.83	4.10	11.39
	Sephacryl S-200 HR 16/60	53,150	0.553	96,112.16	1.04	27.73

**Table 2 foods-12-01820-t002:** The volume, R_g_, hydrophobic SASA, and number of intramolecular hydrogen bonds of sPPO and mPPO according to MD simulation under different pressures.

Pressure (MPa)		Volume (nm^3^)	R_g_ (nm)	SASA (nm^2^)	Hydrogen Bonds
sPPO	0.1	372.242	2.311	12,113.271	185.995
	200	364.894	2.295	11,972.615	186.592
	400	361.495	2.262	11,603.713	180.403
	600	348.742	2.162	12,109.229	171.095
mPPO	0.1	682.572	2.264	16,219.570	406.263
	200	678.115	2.258	16,358.267	411.625
	400	670.246	2.247	16,714.924	409.374
	600	665.864	2.232	16,993.183	402.333

**Table 3 foods-12-01820-t003:** Docking results of sPPO and mPPO with catechol after high-pressure simulation.

	Docking Indexes	0.1 MPa	200 MPa	400 MPa	600 MPa
sPPO	T-Score	4.065	4.696	4.016	3.503
	Number of hydrogen bonds	3	3	2	2
	Amino acid residues involved in hydrogen bonds	His24, His146, His150	Asn76, Trp89, Asn156	His24	Gly209
	Average distance	2.53 Å	2.3 Å	2.75 Å	2.90 Å
mPPO	T-Score	4.196	4.319	3.93	3.866
	Number of hydrogen bonds	3	4	2	2
	Amino acid residues involved in hydrogen bonds	Asn218, Ser246	Arg213, Asp214, Asn218, Ser246	Pro242, Gln455	His93, His116
	Average distance	2.17 Å	2.05 Å	2.60 Å	3.05 Å

## Data Availability

The data presented in this study are available on request from the corresponding author.

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
