# Peer review of "Comparing the Effect of HPP on the Structure and Stability of Soluble and Membrane-Bound Polyphenol Oxidase from ‘Lijiang Snow’ Peach: Multispectroscopic and Molecular Dynamics Simulation"

_foods, 2023, doi:10.3390/foods12091820_

Round 1
Reviewer 1 Report
Comments and Suggestions for Authors
The ms. “Comparing the effect of HPP on the structure and stability of sPPO and mPPO from ‘Lijiang snow’ peach: Multispectroscopic and molecular dynamics simulation” (Ms. Ref. No. foods-2276872-v1) presents original results on the bioinformatics and three-dimensional structures of sPPO and mPPO purified from ‘Lijiang snow’ peaches; the conformational changes of sPPO and mPPO treated by HPP were also investigated. Furthermore, the relationship between enzymatic activities and their structure under high pressure at the molecular level were examined through molecular dynamics simulations combined with multispectral techniques.
There is a lot of work involved and the ms. has evident merit.
The topic falls within the aims and scopes of the Foods journal.
The manuscript is, in general, well written and clearly explained. However, there are a few issues that need to be addressed to improve the presentation of the results.
Major issues:
1. In the Introduction: In my opinion, in the current form of the Introduction (and also in the Abstract), it is not evident that the aim is to reduce the activity of the PPOs to prevent enzymatic browning. I advise to elaborate on this.
2. In the current form, the aim/aims of the work is/are not well addressed. Please first state the hypothesis, then explain the approach.
3. Please discuss how the results of this work and the HPP treatment can be applied in a “real life scenario/case” (i.e. in the processing of fruits), because HPP treatment on intact fruits would result in degradation: is the HPP suitable for purees, or for dry fruits (in powder form), or other form?
4. I advise to use the S.I. units throughout the manuscript. For example, molar concentration should be expressed as mol L-1. The S.I. unit for the enzyme activity is the katal (symbolized as kat).
5. Also regarding the measuring units: in the current form of the manuscript, there is no consistency regarding the volume units, as volume is expresses both in L/mL (e.g. line 100) and in nm3 (Table 2). I suggest using L throughout the manuscript. Please revise.
6. Figure 2: please explain RA in the figure’s caption.
7. Figure 4: please explain the letters in the figure’s caption.
8. Please revise the references and make sure they display all the authors and the identification elements. Apparently, the reference [35] misses authors (denoted as “…”).
9. Please comment your results with respect to similar studies in the literature, such as https://doi.org/10.3390/foods12010167, https://doi.org/10.1016/j.lwt.2022.114156, https://doi.org/10.3390/foods12040905, https://doi.org/10.3390/foods12010221,
10. Please comment the efficiency of the HPP treatment compared to other treatments such as ohmic heating (https://doi.org/10.1016/j.lwt.2021.113021), plasma (https://doi.org/10.3390/foods11213504) or steam explosion treatment (https://doi.org/10.1016/j.lwt.2022.114106) on the PPO activity.
Minor issues:
1. Line 68: exhibited higher resistance.
2. Please revise English throughout the ms. There are several syntax/grammar/typos which need to be fixed.
Given the completed score sheet and the comments above, after careful evaluation, the ms. “Comparing the effect of HPP on the structure and stability of sPPO and mPPO from ‘Lijiang snow’ peach: Multispectroscopic and molecular dynamics simulation” (Ms. Ref. No. foods-2276872-v1) needs Major Revision.
Author Response
1 Question (Q) In the Introduction: In my opinion, in the current form of the Introduction (and also in the Abstract), it is not evident that the aim is to reduce the activity of the PPOs to prevent enzymatic browning. I advise to elaborate on this.
Answer (A): Thank. The introduction and abstract have been revised in the manuscript.
2 Q: In the current form, the aim/aims of the work is/are not well addressed. Please first state the hypothesis, then explain the approach.
A: Thanks for your good suggestion. It has been added in the manuscript. “At present, although numerous studies have investigated on the inhibition of PPO activity by HPP, most of the studies mainly focused only on the inhibition effect of PPO, and few studies explored the difference on the inhibition mechanisms between different forms of PPO at the molecular level. Therefore, in this work, we analyzed the bioinformatics and three-dimensional structures of sPPO and mPPO purified from ‘Lijiang snow’ peaches, investigated the conformational changes of sPPO and mPPO treated by HPP, and further explored the relationship between enzymatic activities and their structure under high pressure at the molecular level through molecular dynamics simulations combined with multispectral techniques. This study can help to better elucidate the inactivation mechanism of HPP on the structure and stability of sPPO and mPPO, and provide a scientific basis for the inhibition of enzymatic browning during the processing and storage of peach products by HPP.”
3 Q: Please discuss how the results of this work and the HPP treatment can be applied in a “real life scenario/case” (i.e. in the processing of fruits), because HPP treatment on intact fruits would result in degradation: is the HPP suitable for purees, or for dry fruits (in powder form), or other form?
A: Thanks. The aim of this study was to investigate the mechanism of HPP on two forms of PPO at the molecular level, elucidate the different pressure resistance of sPPO and mPPO, and provide a theoretical basis for the inhibition of enzymatic browning of PPO by HPP. Thus, sPPO and mPPO were purified from peach to investigate the difference in conformation of two forms of PPO at the molecular level. In food industry, HPP is mainly used in the sterilization of fruit and vegetable juice, fresh cut, and jam, etc. Your comments are very constructive. We now are exploring the effects of HPP and inhibitors on enzymatic browning of sPPO and mPPO in fresh-cut apple slice. According to your suggestion, we will focus our future work on combining the results of this work with the practical application of HPP in fruit and vegetable processing.
4 Q: I advise to use the S.I. units throughout the manuscript. For example, molar concentration should be expressed as mol L-1. The S.I. unit for the enzyme activity is the katal (symbolized as kat).
A: Thanks. It has been revised in the manuscript.
5 Q: Also regarding the measuring units: in the current form of the manuscript, there is no consistency regarding the volume units, as volume is expresses both in L/mL (e.g. line 100) and in nm3 (Table 2). I suggest using L throughout the manuscript. Please revise.
A: Thanks. The units have been revised in the manuscript. In addition, the “nm3” in table 2 referred to the volume of protein, and the unit “nm3” was more appropriate.
6 Q: Figure 2: please explain RA in the figure’s caption.
A: Thanks. It has been added in the manuscript. “The residual enzyme activity (RA) was the percentage of enzyme activity after HPP treatment compared to untreated enzyme activity, which was used to indicate the change of PPO activity after HPP treatment.”
7 Q: Figure 4: please explain the letters in the figure’s caption.
A: Thanks. The letters in the figure were abbreviated as amino acid residues in primary structure of protein. It has been explained in the manuscript as “The letters represented the amino acids that make up the protein.”.
8 Q: Please revise the references and make sure they display all the authors and the identification elements. Apparently, the reference [35] misses authors (denoted as “…”).
A: Thanks. All references have been revised in the manuscript.
9 Q: Please comment your results with respect to similar studies in the literature, such as https://doi.org/10.3390/foods12010167, https://doi.org/10.1016/j.lwt.2022.114156, https://doi.org/10.3390/foods12040905, https://doi.org/10.3390/foods12010221,
A: Thanks very much. Your suggestion is very good, and the literatures you provided have been added in the manuscript. According to your opinion, we have summarized and analyzed the enzymatic properties of the two forms of PPO based on these research findings in the introduction.
10 Q: Please comment the efficiency of the HPP treatment compared to other treatments such as ohmic heating (https://doi.org/10.1016/j.lwt.2021.113021), plasma (https://doi.org/10.3390/foods11213504) or steam explosion treatment (https://doi.org/10.1016/j.lwt.2022.114106) on the PPO activity.
A: Thanks. According to your suggestion, we have added appropriate content in the manuscript for comparative analysis. Based on these studies, we have discussed the characteristics of HPP and thermal processing (including traditional and emerging thermal processing), and summarized the advantages of HPP in the manuscript.
11 Q: Line 68: exhibited higher resistance.
A: Thanks. It has been revised in the manuscript.
12 Q: Please revise English throughout the ms. There are several syntax/grammar/typos which need to be fixed.
A: Thanks. It has been revised in the manuscript.

Reviewer 2 Report
Comments and Suggestions for Authors
It may remain as sPPO and mPPO in the title, but in the abstract it would be better if they were specified as soluble polyphenol oxidase and membrane bound polyphenol oxidase.
In the abstract, the activity of mPPO was stated as 5.3% at 200 MPa, while it was specified as 4.8 to 3.8 on page 6. This should be checked.
In the abstract, it was stated that the residual activity of mPPO decreased by 73% in 50 minutes at 600 MPa, while on page 6 it was stated as 72%.
It will look better if the tables are enlarged. When it is narrow, the words are divided.
Author Response
1 Q: It may remain as sPPO and mPPO in the title, but in the abstract it would be better if they were specified as soluble polyphenol oxidase and membrane bound polyphenol oxidase.
A: Thanks. According to journal IFA, the current title was revised as “Comparing the Effect of HPP on the Structure and Stability of Soluble and Membrane Bound Polyphenol Oxidase from ‘Lijiang Snow’ Peach: Multispectroscopic and Molecular Dynamics Simulation”. It has been revised in the abstract. The changes were as follows: “Polyphenol oxidase (PPO) easily lead to deterioration in the color and nutritional of fruits and vegetables. As a non-thermal process, high pressure processing (HPP) showed different inactivation effect on endogenous enzymes in fruits and vegetables. In this work, soluble PPO (sPPO) and membrane bound PPO (mPPO) from ‘Lijiang snow’ peaches were purified, then the effect of high pressure on the conformation of sPPO and mPPO was investigated and compared at the molecular level.”
2 Q: In the abstract, the activity of mPPO was stated as 5.3% at 200 MPa, while it was specified as 4.8 to 3.8 on page 6. This should be checked.
A: Thanks. There was indeed a mistake here, and it has been revised in the manuscript.
3 Q: In the abstract, it was stated that the residual activity of mPPO decreased by 73% in 50 minutes at 600 MPa, while on page 6 it was stated as 72%.
A: Thanks. We rounded the data in the manuscript previously, and according to your opinion we have modified it and restored it to the actual data.
4 Q: It will look better if the tables are enlarged. When it is narrow, the words are divided.
A: Thanks. It has been revised in the manuscript as your opinion.

Reviewer 3 Report
Comments and Suggestions for Authors
The manuscript entitled “Comparing the effect of HPP on the structure and stability of sPPO and mPPO from ‘Lijiang snow’ peach: Multispectroscopic and molecular dynamics simulation” investigates the effect of high pressure on the conformation of sPPO and mPPO at the molecular level. The authors observed that the maximum activation of sPPO and mPPO by 11.2% and 5.3% after HPP at 200 MPa, while their activities both gradually decreased at 400 MPa and 600 MPa, especially the residual activities of sPPO and mPPO at 600 MPa for 50 min were decreased to 41.4% and 73.0%, respectively. The authors claimed that HPP had little effect on the secondary structure of PPOs, but led to obvious changes in their tertiary structure.Molecular simulations and docking studies have also been reported.
The below revisions are recommended:
- According to the journal IFA, abbreviations are not allowed in the title of a manuscript. Please read the journal IFA and revise the title.
- Please insert a few more refeences to support the statement, “Numerous studies showed that HPP exhibited different inactivation effects on PPO activity depending on resource, and its activity even activated at lower pressure.” (Lines 56-57).
- Please mention the percentage of purity of the ‘purified’ sPPO and mPPO. How the purity was determined?
- Write in detail the procedure used for HPP including the instrumental info.
- Please insert the nESI-LC-MS/MS of sPPO and mPPO chromatograms in the text pointing out (by arrow) the major components.
- Mention the observed difference between the Ramachandran plot of the Psi/Phi distribution of sPPO and mPPO (Fig 1).
- Mention the source of the 3D structure of sPPO and mPPO in the figure caption (Fig 1).
- The manuscript must be thoroughly checked, and the quality of the language must be improved. There are numerous grammatical mistakes.
- Uniformity (font and size) should be mentioned throughout the manuscript, including the schemes and figures. Readability is one of the major criteria for scientific literature. The authors are encouraged to check the journal IFA.
Author Response
1 Q: According to the journal IFA, abbreviations are not allowed in the title of a manuscript. Please read the journal IFA and revise the title.
A: Thanks. It has been revised in the manuscript as your opinion. The current title has been modified as “Comparing the Effect of HPP on the Structure and Stability of Soluble and Membrane Bound Polyphenol Oxidase from ‘Lijiang Snow’ Peach: Multispectroscopic and Molecular Dynamics Simulation”.
2 Q: Please insert a few more refences to support the statement, “Numerous studies showed that HPP exhibited different inactivation effects on PPO activity depending on resource, and its activity even activated at lower pressure.” (Lines 56-57).
A: Thanks. More refences has been added in the manuscript as your suggestion.
3 Q: Please mention the percentage of purity of the ‘purified’ sPPO and mPPO. How the purity was determined?
A: Thanks. We did not determine the specific PPO’s purity by SEC-HPLC. However, the purity of protein samples used for LC-MS/MS analysis is required to be greater than 90%. According to our previous research (Zhou et al., 2023), the purity of PPO was determined by SDS-PAGE and Native-PAGE, especially SDS-PAGE was performed after each purification.
Figure1. SDS-PAGE (a) and Native PAGE (b) of purified sPPO and mPPO.
4 Q: Write in detail the procedure used for HPP including the instrumental info.
A: Thanks. The detail for procedure used HPP and equipment information have been added in the manuscript as your opinion as “In this experiment, a 10 L high-pressure equipment (XC-LF3AH, Jiangmen Xiecheng Machinery Co., Ltd, China) was used to treat sPPO and mPPO, which consisted of a hydraulic booster, a pressure vessel, a pumping system, and a control panel”.
5 Q: Please insert the nESI-LC-MS/MS of sPPO and mPPO chromatograms in the text pointing out (by arrow) the major components.
A: Thanks. After analyzing the identification results, the major component of sPPO was “Polyphenol oxidase II”, while the major component of mPPO was “Tyrosinase_ Cu-bd domain-containing protein”. We have added the Figure S1 as you suggested. Since the sample used for the machine was a peptide segment after protein enzymatic hydrolysis, it was not possible to label the identified protein on the base peak diagram.
Figure 2. Mass spectrum Basepeak diagram of sPPO (a) and mPPO (b).
6 Q: Mention the observed difference between the Ramachandran plot of the Psi/Phi distribution of sPPO and mPPO (Fig 1).
A: Thanks. The Ramachandran plot is a visualization method used to describe whether the dihedral angles ψ and φ of amino acid residues in a protein structure are within a reasonable region, thus reflecting whether the conformation of the protein is reasonable. In many other references, it was common to just discuss the percentage of the allowable region in the Ramachandran plot to indicate whether the protein conformation was reasonable or not, without much detail description of the Ramachandran plot (Liu et al., 2015; Oduselu et al., 2019; Gurung et al., 2020). As you suggested, we have added the following analysis: “In addition, 87.6% and 89.0% of amino acids in sPPO and mPPO were located in the core region (red region), respectively, which might indicate that the conformation of mPPO was more stable than sPPO.”
7 Q: Mention the source of the 3D structure of sPPO and mPPO in the figure caption (Fig 1).
A: Thanks. The source has been added in the manuscript as your opinion.
Q8: The manuscript must be thoroughly checked, and the quality of the language must be improved. There are numerous grammatical mistakes.
A: Thanks. It has been revised in the manuscript as your opinion.
Q9: Uniformity (font and size) should be mentioned throughout the manuscript, including the schemes and figures. Readability is one of the major criteria for scientific literature. The authors are encouraged to check the journal IFA.
A: Thanks. It has been revised in the manuscript as your opinion.

Round 2
Reviewer 1 Report
Comments and Suggestions for Authors
The authors of the revised ms. “Comparing the effect of HPP on the structure and stability of sPPO and mPPO from ‘Lijiang snow’ peach: Multispectroscopic and molecular dynamics simulation” (Ms. Ref. No. foods-2276872-v2) have addressed the reviewers’ comments. Proper changes have been made in the ms. according to suggestions and consequently, the ms. was improved compared to its initial submission.
Therefore, after careful examination, my recommendation term = Accept.
Author Response
Thanks.
Reviewer 3 Report
Comments and Suggestions for Authors
Point#5: Authors have added the MS, which was not wanted for. Please insert the nESI-LC-MS/MS of sPPO and mPPO chromatograms in the text, pointing out (by arrow) the major components.
Author Response
Answer (A): Thanks. Your comment is very good. In this work, the nESI-LC-MS/MS of PPO was carried out by a third-party company (BGI Genomics Co., Ltd) according to the method shown in figure 1. After the experiment completed, the analysis results of two PPOs, including a basepeak diagram, protein identification table, peptide identification table, were provided by the company. Unfortunately, they did not provide us with the secondary mass spectra diagram as usual. Because the secondary mass spectra diagrams were not provided by all publications, we were not aware of the need for a secondary mass spectra diagram. After your reminder, we also communicated with the company, however, our experimental data of the secondary mass spectra diagram was not saved.
As we know, a protein identification table could be generated after confirmation by numerous secondary mass spectra diagrams, and the table was added as figure 2. According to the protein identification result, we identified the main components of the purified sPPO and mPPO were Polyphenol oxidase II & Tyrosinase_ Cu bd domain containing protein respectively, which were both belonged to the category of polyphenol oxidase. Based on your opinion, we will also pay more attention to the secondary mass spectra diagram in protein mass spectrometry identification in future work, and also strengthen the learning knowledge regarding to mass spectrometry.
